# Masticatory Myoelectric Side Modular Ratio Asymmetry during Maximal Biting in Women with and without Temporomandibular Disorders

**DOI:** 10.3390/bios12080654

**Published:** 2022-08-18

**Authors:** Felipe Acácio de Paiva, Kariny Realino Ferreira, Michelle Almeida Barbosa, Alexandre Carvalho Barbosa

**Affiliations:** 1MSc Program in Health Sciences, Federal University of Juiz de Fora, Juiz de Fora 35020-360, Brazil; 2Institute of Health Sciences, Federal University of Juiz de Fora, Juiz de Fora 35020-360, Brazil; 3PhD and MSc Program in Physical Education, Federal University of Juiz de Fora, Juiz de Fora 35020-360, Brazil

**Keywords:** electromyography, facial pain, temporomandibular joint disorders, jaw muscles, diagnosis

## Abstract

There is no consensus on the role of electromyographic analysis in detecting and characterizing the asymmetries of jaw muscle excitation in patients with temporomandibular disorders (TMD). To analyze the TMD patients (*n* = 72) in comparison with the healthy controls (*n* = 30), the surface electromyography (sEMG) of the temporalis anterior muscle (TA) and masseter muscle (M) was recorded while a maximal biting task was performed. The differences in the asymmetry of the relationship between the masseter muscles were assessed in a module to determine the sensitivity (Sn) of binomial logistic models, based on the dominance of the TA or the M muscle, in accurately predicting the presence of TMD. All assumptions were met, and comparisons between the groups showed significant differences for the TA muscle ratio (*p* = 0.007), but not for the M muscle ratio (*p* = 0.13). The left side was predominant over the right side in the TMD group for both the TA (*p* = 0.02) and M muscles (*p* = 0.001), while the non-TMD group had a higher frequency of the right side. Binary logistic regression showed a significant model (χ^2^ = 9.53; *p* = 0.002) for the TA muscle with Sn = 0.843. The model for the M muscle also showed significance (χ^2^ = 8.03; *p* = 0.005) with Sn = 0.837. The TMD patients showed an increased TA muscle ratio and asymmetry of left dominance, compared to the healthy subjects. Both of the binomial logistic models, based on muscle dominance TA or M, were moderately sensitive for predicting the presence of TMD.

## 1. Introduction

The prevalence of temporomandibular disorders (TMD) is 27–38% in the adult population [1]. Chronic TMD affects the patient’s ability to work and interact in the social environment, and results in an impaired quality of life [2,3]. An accurate diagnosis of TMD is critical for treatment planning and increases the chances of successful outcomes. Evidence suggests that individuals with TMD have a lower rate of motor unit discharge in painful muscles [4,5,6] and early fatigue, compared with individuals without TMD [7,8]. Surface electromyography (sEMG) provides a direct and objective assessment of muscle excitation [4,5] and can aid in the diagnosis of TMD, particularly the assessment of masticatory muscle activity in patients with impaired function and altered jaw movement patterns [9,10].

Some studies suggest that the use of sEMG to assess masticatory muscle excitation can discriminate against women with and without TMD [5,11,12,13]. In this sense, some features of muscle functionality in TMD patients have been reported, but the results of specific thresholds for raw electromyography of the masticatory muscles showed poor responsiveness and accuracy in discriminating between healthy and TMD patients [14,15]. One possible solution to this is to compare the excitation asymmetries between the sides and muscles to prioritize intervention [10]. However, no consensus has been reached on the usefulness of sEMG and the presence of muscle excitation asymmetries in TMD patients [14].

Aside from the above-mentioned issues, understanding the musculoskeletal impairments associated with TMD diagnoses is critical to providing effective treatments. Therefore, clarifying the relationship between the excitation of the major jaw muscles between sides during biting could provide insights into managing patients with this complex disorder. In this sense, the primary objective of the present study was to evaluate the difference in the modular relationship between the excitation of the side-to-side muscles by electromyography of the TA and M in TMD patients and healthy subjects, bilaterally. A secondary objective was to evaluate the possible differences, considering a nominal side-to-side predominance in both groups, and to establish possible predictive models with combined sensitivity and specificity analysis. The first hypothesis was that the TMD patients would have an increased asymmetry of ratio and dominance, compared with the healthy subjects. The second hypothesis was that a binomial logistic model, based on the dominance of the TA or M muscle, would be able to accurately predict the presence of TMD.

## 2. Materials and Methods

### 2.1. Participants

Participants (Table 1) were recruited through public invitations, via recruitment posters and personal contacts. A sample of 112 women were interested in participating. The inclusion criterion for the TMD group was a diagnosis of TMD arthralgia that was associated with myofascial pain, according to the Diagnostic Criteria for Temporomandibular Disorders (RDC/TMD), for at least 6 months’ duration. The RDC/TMD is internationally recognized as the gold standard for TMD diagnosis [16]. The assessment included external palpation, using a calibrated pressure algometer (MED.DOR pressure algometry, Governador Valadares, Brazil) [17]. Internal to the mouth, the index finger was calibrated using the pressure algometer to palpate the medial pterygoideus muscles. The inclusion criteria for both groups were having a minimum of 28 permanent teeth and an age between 18 and 45 years. All patients were evaluated by a dentist for periodontal problems. The exclusion criteria for both groups were as follows: a history of trauma to the face and temporomandibular joints; systemic diseases such as arthritis; fibromyalgia; pain due to confirmed migraine; headache or neck pain unrelated to TMD; ongoing use of analgesics, anti-inflammatory drugs, muscle relaxants or psychotropic medications; acute infections or other serious dental, ear, eye, nose, or throat conditions; and neurologic or cognitive deficits. After the initial screening, 10 participants were excluded. The 102 included participants were divided into 2 groups, according to RDC/TMD axis I: 1. The non-TMD group (*n* = 30 individuals without TMD), 2. The TMD group (*n* = 72 individuals with TMD). This cross-sectional study was approved by the Ethics Committee of the Federal College of Juiz de Fora (number 68457617.6.0000.5147). Participants were informed of the benefits and potential risks before signing a written informed consent form, before participating in the study. An a priori sample size was calculated based on a previous study. An effect size of 0.88 was considered (variable: side-to-side electrical excitation %), with α = 0.05 and a power (1-β) of 0.95. The analysis yielded an actual sample power of 0.949, with a total sample size of 60 participants. Considering a drop-out of 20%, the recommended sample included at least 72 participants. G-Power software (version 3.1.5, Franz Faul, College of Kiel, Düsseldorf, Germany) was used to calculate the sample size.

### 2.2. Electromyography

All the sEMG procedures and the device’s specifications were in accordance with the recommendations of the International Society of Electrophysiology and Kinesiology (https://isek.org/, accessed on 25 June 2021). The conversion of analog to digital signals was performed by an A/D board with an input range of 16-bit resolution, a sampling frequency of 2 kHz, a joint-rejection module of more than 100 dB, a signal-to-noise ratio of less than 03 μV root mean square, and an impedance of 109 Ω. The sEMG signals were recorded as root mean square in μV, using surface Meditrace™ (Ludlow Technical Products, Gananoque, ON, Canada) Ag/AgCl electrodes with a diameter of 2 cm and a center-to-center spacing of 2 cm, placed in transverse alignment, parallel to the underlying fibers at a muscle site. Differential bipolar sensors were attached to the electrodes to reduce the constant noise. A reference electrode was placed on the left lateral humeral epicondyle. The sEMG signals were amplified and filtered (Butterworth 4th order, 20–450 Hz bandpass filter, 60 Hz notch filter). All information was recorded and processed using Miotec Suite^®^ software (Miotec Biomedical Equipments, Porto Alegre, RS, Brazil) [7,8,18]. Before placement of the sEMG electrodes, the skin was cleaned with 70% alcohol to remove fatty residues, followed by exfoliation with a special sandpaper for the skin and a second cleaning with alcohol. As the TM and M electrodes’ locations were not described in the Surface Electromyography for the Non-Invasive Assessment of Muscles (SENIAM) site (http://seniam.org/, accessed on 25 June 2021), the electrodes were placed on both the left and right sides, according to previous studies’ descriptions [7,8,18].

### 2.3. Maximal Voluntary Isometric Contraction

The excitation of M and TA muscles was assessed during a maximal bite force test. Each participant performed a 10 s maximal isometric contraction (MVIC), while biting on a load cell (maximum tension–compression = 200 Kgf, precision of 0.1 Kgf, maximum measurement error = 0.33%; Miotec™ Biomedical Equipment, Porto Alegre, RS, Brazil). Subjects were asked to sit comfortably while the adapted arms of the load cell were positioned on the incisors (Figure 1). A disposable material was used to cover the adapted arms for each subject. Forward head posture was controlled during all procedures by positioning the load cell closer to the participant so that participants could bite in their natural head posture. Standardized verbal commands (“begin”, “continue biting”, and “stop”) were used by the same experimenter for all recordings. A 5 s familiarization period was followed by a 3 min pause before the task. The load cell was coupled and synchronized with the electromyograph.

### 2.4. Data Extraction

All data were extracted offline using Miotec Suite™ software (Miotec™, Biomedical Equipments). Because the load cell was synchronized with the electromyography channels, the trained rater determined the interval based on the increase in force. After three 1 s windows of rest were collected, onset was defined by three times the standard deviation from the average rest intervals, plus the mean itself (Figure 2). The interval began when the signal exceeded the onset threshold. Conversely, the end of the interval was defined by the same threshold. For statistical analysis, the mean values of the force intervals were used [7]. The sEMG ratio was calculated using the maximum value (in μV), divided by the minimum value (in μV), so that the difference could always be modular (with positive values). Thus, the difference values were displayed regardless of the side. To determine the sEMG side predominance, a nominal classification was used. If the voltage on the right side had the highest value, the participant’s result was classified as “1”. Conversely, if the left side was higher than the right side, the result was classified as “2”.

### 2.5. Statistical Analysis

The between-group differences were assessed using the independent samples t-tests. The 95% confidence interval was also used to set the lower and the upper limits of significance. The effect size (ES) was set using Cohen’s d coefficient. The magnitude of the ES was qualitatively interpreted using the following thresholds: <0.2, trivial; 0.2–0.6, small; 0.6–1.2, moderate; 1.2–2.0, large; 2.0–4.0, very large; and >4.0, huge [19]. The Chi-square test, with continuity correction, was used to verify the frequency differences for nominal side predominance. To provide a predictive model, binary logistic regressions were performed for M and TA analysis. The assumptions of the absence of multicollinearity (Variance Inflation Factor [VIF] less than 5) and outliers were evaluated. The best fit model was judged based on the values of the Chi-square test, Nagelkerk’s R^2^ and odds ratio (OR), considering its confidence intervals (OR [95% CI]). In addition, the likelihood ratio test, Akaike Information Criteria (AIC), and Bayesian Information Criteria (BIC) values were inspected to assess the model fit (i.e., the lower the better). The sensitivity (Sn) and the specificity (Sp) were also investigated for each model. The correlation between the pain side and the outcome variables were assessed using Pearson’s coefficient (r), accompanied by the adjusted coefficient of determination (r^2^), which is used to measure how well a statistical model predicts an outcome. They were qualitatively interpreted using the following thresholds: <0.1, trivial; 0.11–0.3, small; 0.31–0.5, moderate; 0.51–0.7, large; 0.71–0.9, very large; and >0.9, nearly perfect [19]. All data analysis was performed using the JAMOVI software (v. 1.6.15.0, The JAMOVI Project, 2022), with significance level set at 5%.

## 3. Results

The between-group comparisons showed significant differences for the TA muscle ratio (TMD: 2.17 [1.74] μV vs. non-TMD: 1.50 [1.28] μV; *p* = 0.007; 95% CI: −0.43 to −0.04; ES = 0.35 [small]), but not for the M muscle ratio (TMD: 1.80 [1.46] μV vs. non-TMD: 1.29 [0.23] μV; *p* = 0.13; 95% CI: −0.23 to 0.02; ES = 0.19 [trivial]). In terms of frequency, the left side was predominant over the right side on the TMD group for both the TA muscle (46.6% vs. 31.4%; χ**^2^** = 5.41; *p* = 0.02) and the M muscle (46% vs. 30.9%; χ**^2^** = 11; *p* = 0.001), while the non-TMD group showed a higher frequency of the right side (TA: 13.5% vs. 8.4% and M: 17% vs. 6.4%). The side of pain was only significantly and positively correlated to the temporal ratio index (r = 0.30; *p* = 0.007).

The binary logistic regression showed a significant model (likelihood χ**^2^** = 9.53; *p* = 0.002; Nagelkerk’s R^2^ = 0.119) for the TA muscle, with a 1.07 μV cutoff point. The VIF values confirmed the absence of collinearity (VIF = 1), and the low AIC (=136) and BIC (=141) confirmed the model fit. The model’s Sn was 0.843, with an Sp of 0.431 for an OR of 4.08 (95% CI = 1.599 to 10.400). The M muscle regression model also showed significance, with a cutoff point of 1.11 μV (likelihood χ**^2^** = 8.03; *p* = 0.005; Nagelkerk’s R^2^ = 0.056). The VIF (=1), AIC (=137), and BIC (=142) values confirmed the absence of collinearity and the model fit. The analysis highlighted an OR of 3.64 (95% CI = 1.429 to 9.255), with an Sn of 0.837 and an Sp of 0.415.

## 4. Discussion

The results showed that the TA muscle percentage was two times significantly higher in the TMD group, compared with the non-TMD group. Moreover, the left side was electromyographically predominant in the TMD patients. Conversely, the right side was predominant in the non-TMD participants. The results also showed a predictive value in distinguishing the TMD and non-TMD patients, when considering the TA (OR = 4.08) or the M’s (OR = 3.64) frequency of asymmetry, with a high Sn to detect those with TMD. The primary hypothesis was partially confirmed because differences were observed only in the TA muscle.

Other studies examined the responsiveness of individual variables to distinguish the subjects with TMD from those without TMD. One study examined the pressure–pain threshold (PPT) in 200 subjects of both sexes, aged 19–27 years [20]. Each subject described the result of pressure algometry for the superficial and deep parts of the masseter muscle, the anterior and posterior parts of the temporalis muscle, and the tissues adjacent to the lateral and dorsal parts of the temporomandibular joint capsule, by selecting the pain intensity on a visual analog scale (VAS) each time. A receiver operating characteristic curve analysis showed a specificity of 95.3% in identifying healthy subjects and a sensitivity of 58.4% in identifying patients with TMD symptoms, at a cut-off point of 7.4 VAS and an accuracy of 68.1%. Another study examined a sample of 49 women who were divided into the following three groups: TMJ osteoarthritis, asymptomatic disk displacement, and control group [21]. The authors aimed to determine a cut-off point for PPT and determined the sensitivity and specificity. The specificity determined was 89.6% and the sensitivity was 70%, for a cutoff point of 1.36 kgf/cm^2^ (area under the curve = 0.90). In a previous study, PPT, sensitivity, and specificity were found to be 0.67 and 0.85 for the masseter muscle, and 0.77 and 0.87 for the temporal muscle, respectively, with a cutoff point one standard deviation below the mean PPT of subjects who did not have TMD [20]. Given these results, pressure algometry has severe limitations when used as a single diagnostic tool because of its limited ability to detect true positives (Sn).

The sEMG technique has also been studied, but methodological problems tend to compromise its ability to predict and distinguish the TMD patients from the healthy controls. A previous study showed altered coactivation and coordination strategies of the jaw muscles during mastication, resulting in higher relative energy expenditure and impaired differential recruitment [22]. Another study showed that women with TMD myalgia had greater jaw muscle work than healthy control subjects [9]. However, the same study showed that the activity of the temporalis anterior muscle (TA) and the masseter muscle (M) were similar when comparing the right and left sides in both the TMD and healthy groups, but the TMD group had greater M activity, compared to TA activity. Other studies examined the electromyographic muscle asymmetry between the sides when comparing TMD and healthy subjects [10,13], but the authors reported no differences at rest or during isometric contractions. One study showed the moderate accuracy (0.74–0.84) of raw sEMG in the TA, M, and the suprahyoid muscles in diagnosing TMD at rest, and in the suprahyoid muscles during maximal contraction on parafilm [11]. In addition, the sensitivity ranged from 71.3% to 80% and the specificity ranged from 60.5% to 76.6%. Such conflicting results are often due to different methods of analyzing the electromyographic signal; differences in how TMD patients have been previously diagnosed; and the inclusion of small samples, combined with the lack of sample size calculation. In the present study, the gold standard method for TMD diagnosis was used in conjunction with a large sample size and an a priori calculated power of 0.95. These procedures were performed to ensure further inference on the assessment of asymmetry, instead of the usual raw values for sEMG analysis. Despite the moderate Sp in the current analysis, the main objective was to identify the women with TMD, instead of their healthy counterparts. The analysis yielded values of over 80% sensitivity for the detection of TMD, using nominal side-predominance, with well-established cutoff points. The ideal Sn value is 100%, meaning that all individuals with TMD are detected [23]. However, this value is rarely achieved in clinical or even research studies. Given the possible effect-size bias associated with an inappropriate sample size (type I error), a post-hoc power calculation would demonstrate the value of the results. In the present work, the two-sided post hoc power calculation returned that the power was 0.96 or 96%, with a maximum possible power of 1 or 100% considering the input ES of 0.35, with the sample of 102 participants.

An important issue that must be addressed is the predominance of pain to the left side (57%) over the right side (20%), and over the combination of both sides (23%) for the occurrence of pain (see Table 1). That predominance could be a possible confounding factor that may lead to an interpretation bias of the current results. In fact, the single significant correlation occurred between the pain location and the temporal ratio. However, it was classified as small (r = 0.30; *p* = 0.007), with a trivial coefficient of determination (r^2^ = 0.09). This result means that only 9% of the data fits the predicted model. As a positive correlation, the pain would increase following the increase in the temporal ratio in the TMD patients, but not in an exact linear trend. This means that muscle excitation would also weakly increase proportionally to the pain predominance. The impact of such a left predominance of pain in TMD patients should be further studied, but in the present work its relevance is questionable.

The current study did not aim to establish which side is predominant for all people with TMD, as this may vary according to the location, population age, and other factors to be set in further studies. Instead, the present results reinforce the sEMG as an important factor to account for TMD evaluation, using an alternative index and a nominal classification. As noticed, the imbalance is not restricted to TMD patients, but it is also present (in a different direction) for healthy people. However, the modular ratio showed the intensification of the index for TMD. A limitation of the present study was the cross-sectional design, which did not allow cause–effect inferences. The present study also focused on women in a limited age range. Those with other levels of functional limitations might show a different pattern than their male counterparts of the same age. Not all intrinsic factors were addressed (such as the predominant side for chewing), and the results could be biased by those issues. However, as it also constitutes a memory recall, it was not possible to be sure of such factors for the analysis. Instead, they were pondered by the most usual factors set in previous studies.

## 5. Conclusions

TMD patients exhibit increased TA muscle ratio and an asymmetry of left dominance, compared with healthy subjects. Both binomial logistic models, based on TA or M muscle predominance, were moderately sensitive to predicting the presence of TMD.

## Figures and Tables

**Figure 1 biosensors-12-00654-f001:**
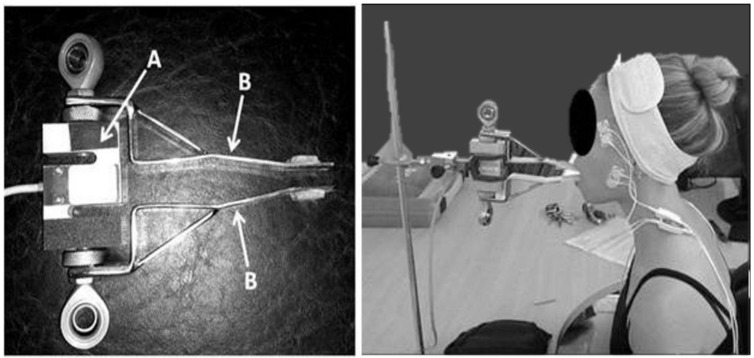
The adapted load cell. (**A**) laboratory-grade load cell; (**B**) adapted arms.

**Figure 2 biosensors-12-00654-f002:**
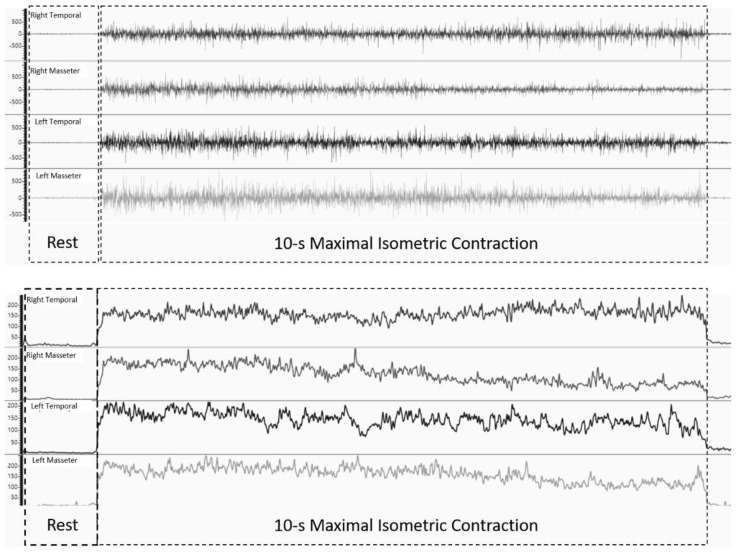
Example of the raw (image above–in μV) and processed RMS signal (image below–in μV).

**Table 1 biosensors-12-00654-t001:** Participants’ Characteristics.

Characteristic		TMD	Non-TMD	*p*
N		72	30	-
Age (years)		29 (6)	29 (4)	0.44
Weight (Kg)		64 (13)	68 (16)	0.18
Height (cm)		163 (6)	164 (5)	0.41
Bite Force (kgf)		13.2 (4.3)	15.1 (2.6)	0.09
Severity (Sample %)	None	0	100	-
	Low	62.5	0	-
	Moderate	37.5	0	-
	Severe	0	0	-
Pain Side Predominance	Left	57%	-	-
	Right	20%	-	-
	Both	23%	-	-

## Data Availability

Data are available only upon a request to the authors.

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
