# Peer review of "Masticatory Myoelectric Side Modular Ratio Asymmetry during Maximal Biting in Women with and without Temporomandibular Disorders"

_biosensors, 2022, doi:10.3390/bios12080654_

Round 1

Reviewer 1 Report

The theme is interesting, however the study design might be compromising the results. The recruiting of the patients did not exclude chewing habits or evaluate any behaviour fator that could be modulating a muscle assimetry. No data over what muscle and whats ide the patients have myalgia, and it could influence the results. Also, in the discussion it was not explain any possible reason for the achieved results.

ABSTRACT:

Needs grammar correction- word: logistic

METHOD:

What type of miofascial pain was diagnosed, with referal pain?

Were patients with Fibromialgia excluded? It could be compromising the results, and some patients presenting TMD also have fibromialgia.

For the diagnosis only clinical evaluation was used, or another tool was employed?

Was there a pattern of muscle pain? The same muscles were involved in all patients? There could be a table showing these details.

Was there chewing on only one side as habit? This information could be influencing the results.

Were there patients with previous or ongoing TMD treament? Did any patient use muscle relaxant to treat muscle pain?

RESULTS:

A table showing the results cound be useful in analysing the data. Also including what muscle in what side did the patient have myalgia.

DISCUSSION:

The discussion is interesting, however did not explain or analyse why patients with TMD have more left muscle assimetry, and control patients have a right side assimetry. It could be due to the confounding factors in selecting patients, and it could be misleading the results.

Author Response

Reviewer #1

The theme is interesting, however the study design might be compromising the results. The recruiting of the patients did not exclude chewing habits or evaluate any behaviour fator that could be modulating a muscle assimetry. No data over what muscle and whats ide the patients have myalgia, and it could influence the results. Also, in the discussion it was not explain any possible reason for the achieved results.

ANSWER: Thank you for your efforts to improve our manuscript’s clarity. Please, see the answers below where we addressed your comments.

ABSTRACT:

Needs grammar correction- word: logistic

ANSWER: Corrected. Thank you for your attention.

METHOD:

What type of miofascial pain was diagnosed, with referal pain?

ANSWER: Thank you for your attention. However, the RDC/TMD has included only one diagnosis of muscle-related disorders for research purposes, that of "myofascial pain with/without limited opening". The DC/TMD, a reviewed version, set the categorization of TMD muscle disorders into the 3 subgroups of local myalgia, myofascial pain with spreading, and myofascial pain with referral. Thus, in the present study, we did not split due to the tool’s limitation.

Were patients with Fibromialgia excluded? It could be compromising the results, and some patients presenting TMD also have fibromialgia.

ANSWER: Thank you for your comment. Yes, the screening included systemic diseases, and we understood the fibromyalgia as one of those, so it was an exclusion criterion. We added the issue in our current version.

For the diagnosis only clinical evaluation was used, or another tool was employed?

ANSWER: Thank you for your comment. Along with the RDC/TMD assessment, there’s muscle palpation. Externally, we used a calibrated pressure algometer, but the internal muscle mouth assessment, we calibrated the finger’s palpation with the algometer. This was not described in our 1st version because it is a part of the RDC/TMD routine. In the present version, we added the information to clarify the procedures.

Was there a pattern of muscle pain? The same muscles were involved in all patients? There could be a table showing these details.

ANSWER: Thank you for your comment. We do have the data and it was included to our characteristics’ table (as it is not an outcome variable). In fact, there was a predominance of pain on the left side. However, and despite the significant correlation, it was interpreted as weak, with trivial coefficient of determination. Please, see the paragraph in our current discussion.

Was there chewing on only one side as habit? This information could be influencing the results.

ANSWER: Thank you for your comment. Unfortunately, we do not have this information. However, this would be based on the patient’s memory recall, which may also impair the results. Thus, we reported it as a limitation.

Were there patients with previous or ongoing TMD treament? Did any patient use muscle relaxant to treat muscle pain?

ANSWER: Thank you for your comment. Previous treatment was not screened. However, no participant was on any ongoing TMD treatment for, at least, 1 year. No medication for any patient. We included the muscle relaxant in our current version.

RESULTS:

A table showing the results cound be useful in analysing the data. Also including what muscle in what side did the patient have myalgia.

ANSWER: Thank you for your comment. However, a table would only repeat the data reported on the text. If this is crucial for readiness, we will be happy to prepare one, but we strongly believe that all reported data is already explained in the text.

DISCUSSION:

The discussion is interesting, however did not explain or analyse why patients with TMD have more left muscle assimetry, and control patients have a right side assimetry. It could be due to the confounding factors in selecting patients, and it could be misleading the results.

ANSWER: Thank you for your comment. However, we need to humbly disagree. The present study’s design cannot establish any cause-effect outcome, but instead analyze the results in the light of previous studies which failed in reinforce the EMG importance for TMD assessment. The present study showed that this is possible using distinct analysis, by splitting the assessment in nominal assessment and index classification. As said, the EMG raw values are very individual specific, and the normalization procedures not always the same (MVIC, peak, rest, etc). In our humble opinion, an alternative analysis would fit better to the patients’ needs, specifically for prospective exercise prescription. Sure, there are limitations in the current study, as any other. However, the biases were avoided by adding more explanation to your previous questioning to reinforce the value of the present results. We tried to make our goals clearer in the corrected discussion.

Reviewer 2 Report

1. Electromyography description is not completed.  For instance, “The conversion of analog to digital signals was performed by an A/D card”. There is no description of A/D card.

2. How was the constant component of the raw signal removed?

3. No examples of raw signals and processed signals. 

4. No examples showing modular ratio asymmetry during maximal biting in women with and without temporomandibular disorders

Author Response

Reviewer #2

  1. Electromyography description is not completed. For instance, “The conversion of analog to digital signals was performed by an A/D card”. There is no description of A/D card.

ANSWER: Thank you for your attention. It is not card, but board. All the specs were according to ISEK recommendations (https://isek.org/wp-content/uploads/2015/05/Standards-for-Reporting-EMG-Data.pdf). However, the EMG description is according to previous published studies (references added) aiming to identify the most important device’s characteristics to acquire EMG data. As told, the Miotec device and the electrodes are in accordance to the ISEK recommendations.  We added more sentences to confirm that info.

  1. How was the constant component of the raw signal removed?

ANSWER: The noise was minimized using bipolar differential sensors. The info was added to our present version.

  1. No examples of raw signals and processed signals.

ANSWER: Thank you for your comment. We added the figure 2 with the raw and processed RMS signals (in microvolts).

  1. No examples showing modular ratio asymmetry during maximal biting in women with and without temporomandibular disorders

ANSWER: Thank you for your attention, but the modular ratio asymmetry was specified in the 1st sentence of the results for both groups: “The between-group comparisons showed significant differences for TA muscle ratio (TMD: 2.17 [1.74] μV vs. non-TMD: 1.50 [1.28] μV; p = 0.007; 95% CI: -0.43 to -0.04; ES = 0.35 [small]), but not for M muscle ratio (TMD: 1.80 [1.46] μV vs. non-TMD: 1.29 [0.23] μV; p = 0.13; 95% CI: -0.23 to 0.02; ES = 0.19 [trivial]).” We are not sure what the reviewer meant as “examples”. This is how the ratio was calculated: “The sEMG ratio was calculated using the maximum value (in μV) divided by the mini-mum value (in μV), so that the difference could always be modular (with positive values).”

Round 2

Reviewer 2 Report

no more comments